

# Intestinal GPDH-1 regulates high glucose diet induced lifespan extension in aged worms

Jihao Mo[1,*], Zhenzhen Zhang[2,*], Xiaowei Wang[3], Miaomiao Wang[4], Ning Sun[5], Lei Wang[3] and Meimei Wang[3]

[1] Luoyang Orthopedic Hospital of Henan Province, Henan, China
[2] Shenzhen Institute of Advanced Technology, Shenzhen, China
[3] Anhui Medical University, Hefei, China
[4] Huang He Science and Technology University, Zhengzhou, China
[5] Nanjing University, Nanjing, China
* These authors contributed equally to this work.

## ABSTRACT

A high glucose diet (HGD) is associated with many metabolic diseases including type 2 diabetes, and cardiovascular diseases. Additionally, a HGD increases the oxidative stress resistance of young animals but shortens their lifespan. To investigate the role of HGD feeding on the aging of aged animals, we tested for oxidative stress resistance and changes in lifespan using *C. elegans*. We showed that a HGD extends the lifespan of aged worms that are dependent on oxidative stress resistance. Furthermore, we measured the lifespan of oxidative stress responding genes of HGD-fed worms. We found that *gpdh-1* and *col-92* are highly expressed in HGD and paraquat (PQ) treated worms. Further experiments indicated that intestinal *gpdh-1* is essential for the HGD induced lifespan extension of aged worms. Our studies provide new insights into understanding the correlation between glucose metabolism, oxidative stress resistance, and aging.

# INTRODUCTION

Aging is one of the most critical features of many metabolic diseases including type 2 diabetes, cardiovascular diseases, and neurodegenerative diseases (*Spinelli et al., 2020*; *Gunasekaran & Gannon, 2011*; *Rodgers et al., 2019*; *Hou et al., 2019*). The factors that regulate aging include both genetic and environmental factors (*Rodriguez-Rodero et al., 2011*). Diet is one of the most well-studied controllable environmental factors that affect aging. Calorie restriction was found to increase lifespan firstly in rodents, then this was carried out in *C. elegans*, *Drosophila*, primates, and other organisms (*Lakowski & Hekimi, 1998*; *Partridge, Piper & Mair, 2005*; *Roth, Ingram & Lane, 1999*; *Zhang et al., 2019*). Lower calorie intake reduces the metabolic rate and produces fewer free radicals, which decreases the risk to get metabolic diseases as there are less DNA and protein damage (*Gredilla & Barja, 2005*). Other studies investigated the effect on longevity using the same calorie intake but changing the components of food, such as a high fat diet (HFD) and high

Corresponding authors
Lei Wang, hantong2046@sohu.com
Meimei Wang, wangmm@ustc.edu.cn

glucose diet (HGD) (*Hariri & Thibault, 2010*; *Alcantar-Fernandez et al., 2018*). A HFD and a HGD increase body weight and induces obesity (*Hariri & Thibault, 2010*; *Alcantar-Fernandez et al., 2018*). A correlation between HFD and cognitive impairment has been established (*Kothari et al., 2017*). Although previous research built a strong association between calorie control and lifespan, the mechanism of food intake and lifespan remains unclear.

The free radical theory believes that aging is caused by the accumulation of oxidative damage over time (*Harman, 1992*). The amplification of free radicals is highly active and harmful (*Jenkins, 1988*). These molecules can damage DNA, RNA, protein, and fatty acids, which eventually accumulates injuries that cause age-related diseases (*Phaniendra, Jestadi & Periyasamy, 2015*). Based on this theory, *in vivo* free radicals are generated by metabolic processes, so slowing down the metabolism rate or enhancing the oxidative stress resistance should extend lifespan (*Salmon, Richardson & Perez, 2010*). It is consistent with the findings that antioxidant supplementation such as vitamin E, polydatin, or quercetin extends lifespan (*Ishii et al., 2004*; *Wen, Gao & Qin, 2014*; *Surco-Laos et al., 2011*). In contrast, knockdown of the oxidative stress response transcription factor *skn-1* shortens lifespan (*Grushko et al., 2021*). However, when adult worms are fed a high glucose diet from Day 1, they have increased oxidative stress resistance and a shortened lifespan (*Lee, Murphy & Kenyon, 2009*; *Wang et al., 2019*). But the knockout of oxidative stress related transcription factor *skn-1* reversed that shortened lifespan (*Lee, Murphy & Kenyon, 2009*). The mechanism of how HGD regulates oxidative stress resistance and lifespan needs further investigation.

Recent studies showed that the developmental stage of HGD feeding is one of the crucial factors that are important for lifespan regulation (*Lee, Murphy & Kenyon, 2009*; *Lei, Beaudoin-Chabot & Thibault, 2018*). The HGD feeding from larva stage 1 or adult Day 1 shortens the lifespan of *C. elegans* (*Lee, Murphy & Kenyon, 2009*). However, a HGD from the post-developmental stage increases the movement of aged worms and decreases the bacteria food intake (*Lei, Beaudoin-Chabot & Thibault, 2018*). To investigate the relationship between aging and oxidative stress resistance in HGD-fed worms, we used paraquat (PQ) to measure the oxidative stress resistance in HGD-fed worms at pre- and post-reproductive stages. By analyzing the RNA sequencing data, we found four gene candidates of HGD-feeding changed oxidative stress genes. Then, we tested the oxidative stress resistance and lifespan in these mutant worms. Our findings will shed new light on understanding the relation between glucose metabolism, oxidative stress resistance, and aging.

## MATERIALS AND METHODS

### *C. elegans* strains and culture conditions

The strains of *C. elegans* were gifted by the Caenorhabditis Genetics Center (St. Paul, MN, USA) and were cultured as previously described (*Brenner, 1974*). Briefly, the worms were maintained on nematode growth medium (NGM) plates at 20 °C with *E. coli OP50*. The HGD plates were made by adding 2% glucose to the normal NGM medium.

To eliminate the possibility of HGD affecting the growth of *OP50*, autoclave-killed *OP50* was used as a food source in both control and HGD plates.

## RNAi strains and culture conditions

The RNAi worms were generated by feeding following a standard procedure as described (*Conte et al., 2015*). The *E. coli* HT115 expressing empty vector *L4440* was used as a control and *HT115* expressing *col-92*, *col-94*, or *gpdh-1* genome DNA fragment was used as food for relative RNAi conditions. Briefly, the *col-92*, *col-94*, and *gpdh-1* RNAi bacteria were picked and cultured on LB plates with 1 mM ampicillin and 1 mM tetracycline. After that, the bacteria were cultured at 37 °C overnight. Then the bacteria were seeded on NGM plates with 1 mM IPTG. Finally, newly synchronized embryos of wild-type worms were seeded on these plates for experiments.

The intestinal and neuronal *gpdh-1* RNAi strains were generated by micro-injection as described (*Calixto et al., 2010*). Briefly, the 5 kb *elt-2* promoter was used for intestinal specific expression, and the 3.5 kb *unc-119* promoter was used for neuronal specific expression of *sid-1* cDNA with unc-54 UTR in *sid-1 (pk3321)* mutants. Then the vectors were purified and injected with an *unc-122*::GFP marker respectively. Then the vectors were purified and injected with an *unc-122*::GFP marker respectively. The GFP expressed worms were fed with *col-92*, or *gpdh-1* RNAi bacteria for the lifespan assay respectively.

## Lifespan assay

The lifespan was measured following a standard procedure with minor modifications (*Wang et al., 2015*). Briefly, newly synchronized embryos of wild-type worms were seeded on NGM plates containing *OP50* as food. Twenty worms were transferred to each plate using an autoclaved eyebrow hair and there were five plates for each group. The first day of adulthood was defined as Day 1. The worms were transferred every 24 h by an autoclaved eyebrow hair during the egg-laying stage. Then the adult worms were transferred every 2 days till death. The worms that ceased pharyngeal pumping and had no response to gentle touch by the eyebrow hair were recorded as dead. The final concentration of PQ used for the lifespan assay was 0.2 mM. *P*-values were analyzed with the Log-rank (Mantel-Cox) test using survival analysis from GraphPad 8.3.1. The lifespan rawdata and statistics are listed in Tables S1 and S2.

## RNA-seq data analysis

The RNA sequencing data (GSE123531, GSE182981, GSE54024) was sourced from the Gene Expression Omnibus, National Center for Biotechnology Information. By using package ggplot2, the volcano plot was generated through R (version 3.5.2; *R Core Team, 2018*; *Wickham, 2016*). The genes marked red showed two-fold changes and have a *P*-value < 0.05. The genes marked cyan showed two-fold changes and have a *P*-value > 0.05. The Database for Annotation, Visualization and Integrated Discovery (DAVID v6.8) was used to process the analysis of gene function (*Dennis et al., 2003*). Gene functional classification was done using the significantly decreased and increased gene lists, separately. The detail of the gene functional classification was attached in Table S3.

**Table 1 The detail of primers.** The sequence of primers.

| Primer | Sequence |
|--------|----------|
| *gpdh-1* sense RNAi F | GGCGGATCCCTGACAACGTC |
| *gpdh-1* sense RNAi R | CGCCTCGCAGAACTTCCCAT |
| *gpdh-1* antisense RNAi F | ATGGGAAGTTCTGCGAGGCG |
| *gpdh-1* antisense RNAi R | GACGTTGTCAGGGATCCGCC |
| *gpdh-1* qPCR F | GGCGGATCCCTGACAACGTC |
| *gpdh-1* qPCR R | CGCGTGGGCTCCCTTTTGTA |
| *col-92* qPCR F | CGCTGTGCATCACCCTTCCA |
| *col-92* qPCR R | GGGATCCGGTGTTTCCGGTC |
| *col-94* qPCR F | TTCCCGTTTCGCCCGTCAAG |
| *col-94* qPCR R | GGTGGTGGGGTGATTGGCTC |
| *ftn-1* qPCR F | CGTGGAGGACGTGTTGCCAT |
| *ftn-1* qPCR R | TGCGCGTCATTGCGTTGTTC |

## mRNA expression

Quantitative real-time PCR (qPCR) screening was performed as previously described with some modifications (*Lei et al., 2017*). About 1,000 synchronized worms were harvested after their cultivation on NGM plates for 3 or 7 days. With a motorized pestle homogenizer, the worms were lysed. TRIzol reagent (Thermo Fisher, Waltham, MA, USA) was used to isolate the total RNA. Following the manufacturer's protocol, TURBO DNase (Thermo Fisher, Waltham, MA, USA) was used to remove the contaminant DNA from the samples. RevertAid Reverse Transcriptase (Thermo Fisher, Waltham, MA, USA) was used to synthesize complementary DNA (cDNA) from the total RNA. By using a Real-time PCR system CFX-96 (Bio-Rad, Hercules, CA, USA), qPCR was carried out with SYBR Green (Qiagen, CA, USA) following the manufacturer's protocol. A total of 50 nM of the paired primer for target genes and 30 nanograms of cDNA were adopted in each reaction. The relative abundance of mRNA was normalized to *act-1* as the housekeeping gene. The *act-1* primers used in this assay are the same as previously reported (*Lei, Beaudoin-Chabot & Thibault, 2018*). Other primers used for the qPCR are listed in Table 1.

## Oxidative stress resistance assay

The resistance to PQ was measured as previously described with minor modifications (*Wang et al., 2019*). Briefly, worms were allowed to grow on agar plates with ND until they were either Day 1 or Day 5 adults. Then, the worms were transferred to ND or HGD plates with or without 125 mM PQ. The alive worms were counted every 2 h till all the worms died. *P*-values were analyzed using Log-rank (Mantel-Cox) test using survival analysis from GraphPad 8.3.1. The oxidative stress resistance raw data was listed in Table S4.

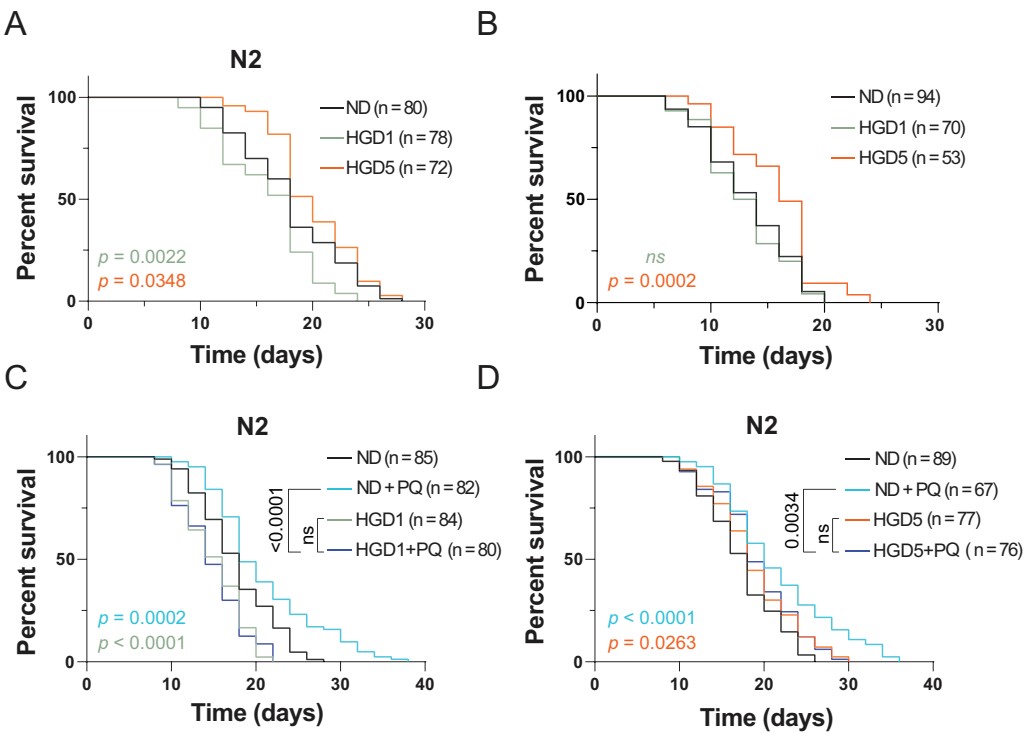

**Figure 1 Paraquat eliminates the lifespan extension effect of HGD on aged worms.** The lifespan assay of (A) wild type N2, and (B) *skn-1* knockout mutants, on HGD Day 1 and Day 5 adults. (C) The lifespan assay of PQ treated mutants on HGD Day 1 worms. (D) The lifespan assay of PQ treated mutants on HGD Day 5 worms. ND, normal diet. HGD1, high glucose diet starts from Day 1 adults. HGD5, high glucose diet starts from Day 5 adults. PQ, paraquat. The number of worms and *P*-value were indicated in the figures. The *P*-value was calculated by the Log-rank (Mantel-Cox) test.

## RESULTS

### Paraquat eliminate the lifespan extension effect of HGD on aged worms

HGD feeding, starting from the egg or Day 1 adult worms, has been known to shorten lifespan and enhance oxidative stress resistance for decades (*Lee, Murphy & Kenyon, 2009*). The lifespan regulation by HGD feeding in young worms is regulated through the oxidative stress transcription factor SKN-1 (*Lee, Murphy & Kenyon, 2009*). However, recent studies showed that HGD feeding starting from the post-developmental stage extends lifespan in aged worms (*Lei, Beaudoin-Chabot & Thibault, 2018*). These results suggested that the mechanisms of HGD feeding on longevity in the pre- and post-developmental stages are different. To test if SKN-1 regulates the lifespan in HGD-fed aged worms, we measured the lifespan of *skn-1* knockout mutants by HGD feeding starting from Day 1 (HGD1) adults and Day 5 adults (HGD5). As shown in Figs. 1 A and 1B, HGD feeding starting from Day 5 adults extended the lifespan in *skn-1* mutants just like in wild type worms. These results suggested that the lifespan regulation in HGD-fed aged worms was independent of SKN-1.

PQ is a chemical that is widely used to induce oxidative stress in animals. High concentrations of PQ shortens the lifespan in *C. elegans* but lower concentrations of PQ extends the lifespan (*Ji et al., 2022*). This phenomenon is due to the high concentration of oxidative stress causing damage to DNA, RNA, and proteins which induces cell death. But a lower concentration of PQ increases oxidative stress resistance which positively regulates lifespan in animals (*Gusarov et al., 2017*). Additionally, *Wang et al. (2019)* reported that PQ eliminates the lifespan shortening effect in young worms. A possible explanation is that a lower concentration of PQ increases the oxidative stress resistance which helps the worms to overcome the HGD induced stresses during the aging process. To test the effect of PQ on HGD induced lifespan extension in aged worm, we tested the lifespan by using Day 5 adults exposed to HGD and PQ. Our data showed that PQ eliminates HGD induced lifespan extension in the aged worm (Figs. 1C and 1D). These results suggested that PQ-specific responding genes eliminate the effect of HGD specific responding genes which positively regulates lifespan in aged animals.

### *Gpdh*-1 and *col-92* are the PQ and HGD target genes in aged worms

To elucidate the mechanism of PQ and HGD on lifespan regulation, we analyzed the target genes for HGD-fed Day 5 adult worms, PQ, and SKN-1 target genes (*Grushko et al., 2021*; *Lei, Beaudoin-Chabot & Thibault, 2018*; *Yee, Yang & Hekimi, 2014*). There are 20 genes significantly up-regulated, and 11 genes significantly down-regulated that are in common between HGD5 and PQ treated conditions, independent of SKN-1 (Fig. 2A).

To understand the function of these genes, we proceeded with tissue and gene function enrichment analysis. The down-regulated genes are intestine and excretory duct cell enriched, while the up-regulated genes do not have enriched tissues. The gene function analysis results show that the up-regulated genes are characterized as being organism oxidative stress resistant, causing an extended lifespan, has oxidoreductase activity acting on CH-OH group of donors, and so on (Fig. 2B). We hypothesize that the oxidative stress and lifespan related genes are the target genes that regulate lifespan. In aged worms treated with HGD and PQ, the target genes *gpdh-1* and *col-92* have increased mRNA expression compared to the PQ or HGD exposure group (Figs. 2C–2F). The GPDH-1 is a glycerol-3-phosphate dehydrogenase that catalyzes the conversion of dihydroxyacetone phosphate to glycerol 3-phosphate (G3P) and is mainly expressed in the intestine and neurons of *C. elegans*. COL-92 is a collagen gene expressed in the hypodermis. Previous findings reported that the knockdown of *gpdh-1* or *col-92* shortens the lifespan of worms (*Iatsenko et al., 2013*; *Possik et al., 2022*), which indicated that these two genes are crucial for lifespan regulation. To conclude, *gpdh-1* and *col-92* are the target genes for HGD or PQ treatment which are related to enhanced oxidative stress and lifespan extension. They are highly expressed in HGD and PQ treated worms.

### *Gpdh*-1 is regulating the PQ induced oxidative stress and lifespan in HGD-fed aged worms

To further elucidate the function of *gpdh-1* and *col-92* on oxidative stress resistance of aged worms, we measured the lifespan of adult worms under high concentrations of PQ.

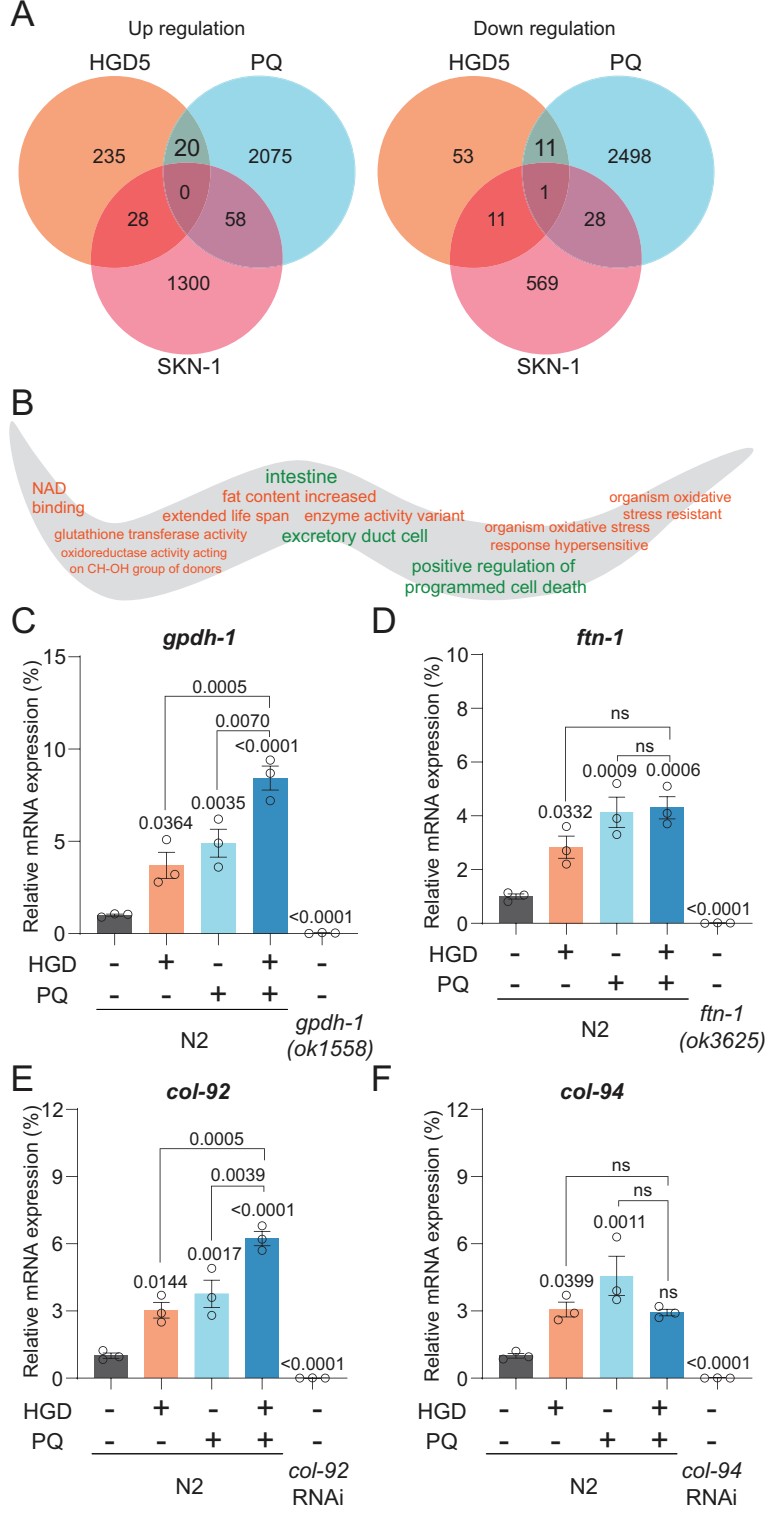

**Figure 2** *Gpdh-1* and *col-92* are highly expressed in HGD-fed aged worms. (A) The Venn diagram of significantly up- and down-regulated genes of HGD5 (orange), PQ (purple), and *skn-1* (blue) knockout groups. (B) The gene ontology analysis of the enriched genes for HGD5 and PQ groups but not *skn-1* dependent. The enriched terms for up-regulated genes were labeled orange, and for down-regulated genes were labeled green. (C–F) The mRNA expression of *gpdh-1*, *ftn-1*, *col-92*, and *col-94* for wild type

**Figure 2** (continued)
worms with and without HGD and PQ treatment. For *gpdh-1* and *ftn-1*, the knockout worms were used as a positive control. For *col-92* and *col-94*, the knockdown worms were used as a positive control. The P-value was calculated by one way ANOVA.  

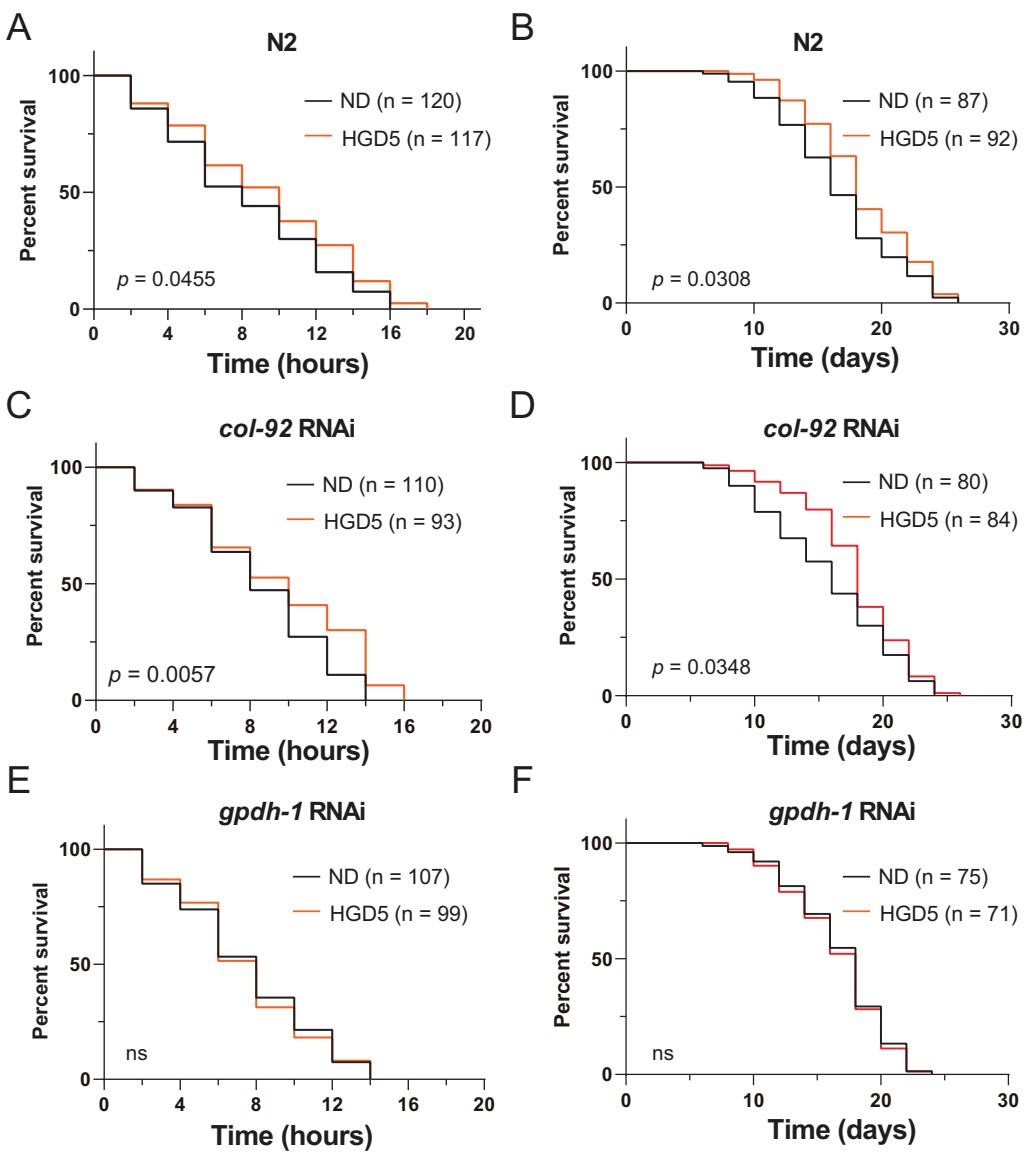

**Figure 3** (A-F) **The HGD5 elevated oxidative stress resistance and lifespan extension in animals were dependent on *gpdh-1*.**  

The HGD5 worms survived longer than the control worms (Figs. 3A and 3B). This is consistent with the studies in HGD-fed young worms. The *col-92* RNAi worms behave similarly to wild type worms (Figs. 3C and 3D). Notably, knockdown of *gpdh-1* eliminates the oxidative stress resistance and the extended lifespan effect of HGD feeding in aged worms (Figs. 3E and 3F). But increased oxidative stress resistance might not always lead to lifespan extension. For example, HGD feeding from Day 1 adults increases oxidative stress

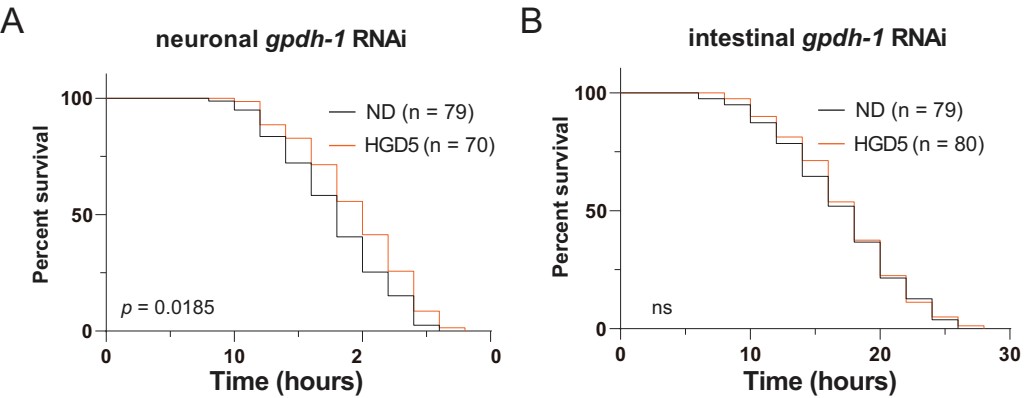

**Figure 4 Intestinal GPDH-1 is required for HGD5 induced lifespan extension in aged worms.** The lifespan assay of (A) neuronal *gpdh-1* RNAi, (B) intestinal *gpdh-1* RNAi, on HGD5 adults. The number of worms and *P*-value were indicated in the figures. The *P*-value was calculated by the Log-rank (Mantel-Cox) test.

resistance but caused a shortened lifespan. To study the role of *gpdh-1* and *col-92* on HGD-fed aged worms, we further measured the lifespan of Day 5 adults in *col-92* and *gpdh-1* knockdown strains. The HGD-fed Day 5 adult worms had extended lifespans in the empty vector and *col-92* RNAi groups. No significant difference was found between the control and HGD5 groups for *gpdh-1* knockdown worms (Figs. 3E–3G). These results showed that *gpdh-1* is required for oxidative stress resistance and lifespan regulation in aged worms.

## Intestinal *gpdh-1* is required for lifespan extension in HGD-fed aged worms

Since *gpdh-1* is mainly expressed in intestinal cells and neurons, we generated intestinal and neuronal specific knockdown strains to investigate the function of *gpdh-1* on lifespan extension in HGD-fed aged worms. Injection of dsRNA or dsDNA might spread to nearby cells and organs, so here we used the systemic RNA Interference defective stain *sid-1* (*pk3321*). Then we created a neuronal specific rescue of SID-1, and an intestinal specific rescue of SID-1 in these *sid-1* defective mutants. After that, we fed them with the *col-92* or *gpdh-1* RNAi bacteria, to promote knockdown in each tissue, respectively. Neuronal knockdown of *gpdh-1* showed increased lifespan in HGD-fed aged worms (Fig. 4A). But no difference was found by when comparing control and HGD-fed aged worms in the intestinal knockdown worms (Fig. 4B). To conclude, the DAF-16 and SKN-1 pathways were activated in Day 1 adult worms but not day 5. The PQ activated oxidative stress (OS) was needed for lifespan regulation in both Day 1 and Day 5 adult worms. In Day 5 worms, the PQ activated OS increased intestinal expression of *gpdh-1*, which helps the HGD-fed aged worms to live longer (Fig. 5).

## DISCUSSION

The insulin pathway was reported as the major regulator for lifespan regulation in HGD-fed Day 1 adult worms (*Lee, Murphy & Kenyon, 2009*). DAF-16 knockout mutants

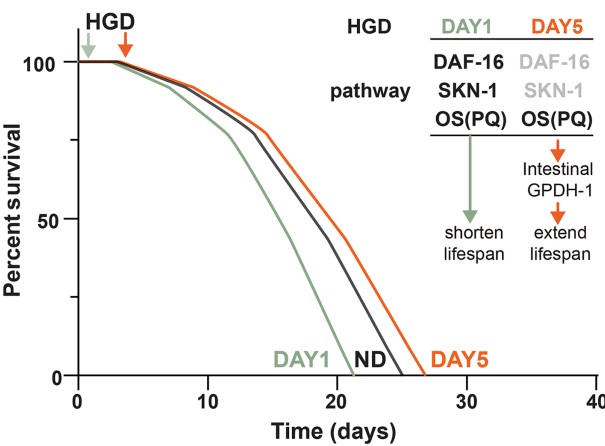

**Figure 5 Schematic diagram of how HGD regulates worms' lifespan.** When Day 1 adult worms were fed a HGD, they have a shortened lifespan compared to those with a normal diet. The lifespan regulation of Day 1 adult worms fed with a HGD was dependent on DAF-16, SKN-1, and PQ activated oxidative stress (OS). When Day 5 adult worms were fed with a HGD, they have an extended lifespan. The lifespan regulation of Day 5 adult worms fed with a HGD was dependent on PQ activated OS. The DAF-16 and SKN-1 pathways were not activated in Day 5 adult worms. The PQ activated OS increased intestinal expression of *gpdh-1*, which led to an extended lifespan.

eliminated the HGD induced lifespan shortening effect in those worms (*Lee, Murphy & Kenyon, 2009*). While the activation of DAF-16 decreases with age (*Baxi et al., 2017*), the transcription factor DAF-16 activation was attenuated in the post-developmental stage worms when fed with HGD (*Lei, Beaudoin-Chabot & Thibault, 2018*). This is consistent with the result that Day 5 adult *daf-16* mutants have an extended lifespan when fed a HGD (*Lei, Beaudoin-Chabot & Thibault, 2018*). These results indicate that the insulin pathway loses control of high glucose caused stresses when animals are old.

The previous studies found that the lifespan extension of HGD5 on worms were dependent on UPR transducer *pek-1* and *atf-6* (*Lei, Beaudoin-Chabot & Thibault, 2018*). The *pek-1* and *atf-6* pathways were regulating protein homeostasis through the degradation of damaged/misfolded protein and regulating the mRNA expression level of UPR genes in animals (*Koh et al., 2018*). Oxidative stress could cause protein damage or misfolding (*Ezraty et al., 2017*). Indeed, the accumulated oxidative stresses might also damage lipids, DNA, and RNA (*Markesbery & Lovell, 2007*). In aged worms, the damaged DNA and lipids take a longer time to react compared to damaged proteins, especially enzymes, due to the reduced cell division and metabolic rates. Thus, protein homeostasis was disrupted more severely in aged animals. Furthermore, Todd et al. reported that damaged proteins serve as a signal that activates *gpdh-1* expression and glycerol synthesis (*Lamitina, Huang & Strange, 2006*). A possible explanation might be that in the aged animals fed with a HGD, the oxidative stress damages protein which activates UPR transducer *pek-1* and *atf-6*, and *gpdh-1* expression.

Accumulated damage from oxidative stresses are one of the major causes of aging. However, either an increase or decrease in oxidative stress resistance might not always come with an lifespan extension or reduction. For example, superoxide radicals ($O_2^-$) have been considered a major cause of aging and serves as strong evidence that supports the

oxidative damage theory of aging. However, manipulating five sod genes in *C. elegans*, which detoxify superoxide radical, has little or no effect on lifespan. But overexpressing *sod-1* does extend lifespan (*Doonan et al., 2008*). The lifespan extension was caused by activating unfolded protein response transducer IRE-1, longevity-promoting transcription factors HSF-1 and DAF-16 rather than the of removal $O_2^-$ (*Cabreiro et al., 2011*). Overall, oxidative damage is just one of the several specific damage types that contributes to lifespan regulation (*Gladyshev, 2014*). Our results suggest that HGD-feeding increases oxidative stress resistance in both young and aged worms. The increased oxidative stress resistance helps the aged worms to counter aging through intestinal GPDH-1 expression, but this did not help the young worms live longer. The GPDH-1 is known for producing high amounts of protective osmolyte glycerol within the intestine. In the recent publication, they found that sorbitol treatment extends lifespan in a *gpdh-1*/*gpdh-2* dependent manner (*Chandler-Brown et al., 2015*). In our hypothesis, overexpression of *gpdh-1* will cause more osmolyte glycerol accumulation, which should be similar to sorbitol supplementation, and further extends lifespan. Further experiments should focus on the effect of oxidative stress resistance in HGD-fed young worms. These experiments will shed new light on understanding the relationship between glucose metabolism, oxidative damage, and aging.

### Funding

This work was supported by grants from the National Natural Science Foundation of China (22176002); the Young Scientists Fund of the National Natural Science Foundation of China (82201315); the Anhui Provincial Natural Science Foundation (2008085MB49); Open Project Fund of the Key Laboratory of the Ministry of Education for the Birth Population (JKZD20202); the Funded Project of Anhui Medical University's Research Level Improvement Program (2021xkjT004) and the Funded Project of Anhui Medical University Basic and Clinical Cooperative Research Promotion Program (2021xkjT041). The funders had no role in study design, data collection and analysis, decision to publish, or preparation of the manuscript.

### Grant Disclosures

The following grant information was disclosed by the authors:
National Natural Science Foundation of China: 22176002.
Young Scientists Fund of the National Natural Science Foundation of China: 82201315.
Anhui Provincial Natural Science Foundation: 2008085MB49.
Key Laboratory of the Ministry of Education for the Birth Population: JKZD20202.
Anhui Medical University's Research Level Improvement Program: 2021xkjT004.
Anhui Medical University Basic and Clinical Cooperative Research Promotion Program: 2021xkjT041.

## Competing Interests

The authors declare that they have no competing interests.

## Author Contributions

- Jihao Mo conceived and designed the experiments, performed the experiments, analyzed the data, prepared figures and/or tables, authored or reviewed drafts of the article, and approved the final draft.
- Zhenzhen Zhang performed the experiments, analyzed the data, prepared figures and/or tables, authored or reviewed drafts of the article, and approved the final draft.
- Xiaowei Wang performed the experiments, analyzed the data, prepared figures and/or tables, authored or reviewed drafts of the article, and approved the final draft.
- Miaomiao Wang performed the experiments, authored or reviewed drafts of the article, and approved the final draft.
- Ning Sun performed the experiments, authored or reviewed drafts of the article, and approved the final draft.
- Lei Wang conceived and designed the experiments, authored or reviewed drafts of the article, and approved the final draft.
- Meimei Wang analyzed the data, prepared figures and/or tables, authored or reviewed drafts of the article, and approved the final draft.

## Data Availability

The raw data are available in the Supplemental Files.

## Supplemental Information

Supplemental information for this article can be found online at http://dx.doi.org/10.7717/peerj.15845#supplemental-information.

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
