# Peer review of "Intestinal GPDH-1 regulates high glucose diet induced lifespan extension in aged worms"

_PeerJ, doi:10.7717/peerj.15845_

## Round 0.1 · original submission · Major Revisions

Both reviewers are a little underwhelmed by this study and have raised concerns which you should consider. Both comment on the clarity of writing which needs careful attention.

Please address the comments fully and make the suggested changes to your manuscript. In particular, address the relationship of the RNAseq datasets used to the question posed, explain the statistics and the method used for intestinal rRNAi, outline the approach used regarding the media and consider performing the experiment on over expression. I feel the latter is needed to validate these studies.

Reviewer 1 ·

Basic reporting

This paper tries to demonstrate that intestinal GPDH-1 regulates an increase in lifespan in worms fed high-glucose diets. Unfortunately, it is difficult to follow the experimental logic behind it. Furthermore, the English language should be improved to better understand it.

Experimental design

They used three different sets of results of previosuly reported RNA-Seq experiments (one for skn-1 mutants, one for high-glucose diet and another for oxidative stress), but it is not justified why they selected and used these results and how they contribute to answer their research question.

They performed RNAi experiments (fig. 4), but no single note on RNAi procedures is given in the Material and Methods section.

Validity of the findings

Even when the research goal is very limited in scope, the results do not firmly fill a research gap. Conclusions are not validated by the results.

Reviewer 2 ·

Basic reporting

Previous studies already reported that high glucose diet shortens C. elegans lifespan if starting from egg but extends lifespan if starting during adulthood. This study aims to identify the downstream genes that may underlie the pro-longevity effect of HGD in adult worms. They showed that GPDH-1 is such a gene. Although this finding is useful for the field, there are quite a few shortcomings in this study, which will be elaborated in below sections. These pitfalls have significantly dampened my overall enthusiasm.

Experimental design

1) No statistics table is presented in this study, which makes it difficult to judge how significant the differences are. For example, ND vs HGD5 in Fig 1A, the difference in lifespan is rather small to me
2) how was intestine-specific RNAi done? Should it be done in the sid-1 mutant background? Otherwise dsRNA will be transported out of the intestine and cause systemic RNAi anyway.
3) does gpdh-1 overexpression extend lifespan?
4) It is difficult to grow C. elegans on dead OP50 due to the lack of vitamin B2. But authors used autoclaved OP50 as the food source for all their studies. How?

Validity of the findings

Critiques outlined in section 2 need to be addressed to validate the findings.

Additional comments

The quality of writing is rather poor, with a lot of errors and grammar mistakes. A thorough proofreading is needed during revision. Two paragraphs starting from line 194 and 207 seem identical to me? Which reflects the negligence of the authors.

---

## Round 0.2 · Minor Revisions

I am satisfied with your responses to the issues raised with one exception. While I appreciate the issues with over-expression and understand the experimental difficulties, I think this point and your argument in your rebuttal letter should be placed into the discussion of the paper so that readers can be aware of this issue.

Could you kindly amend the discussion to reflect this?

I am then minded to recommend acceptance.

Thanks for your considered responses and appropriate changes to the text.

---

## Round 0.3 · accepted · Accept

Thanks for dealing with the final point. Congratulations.